# PRMT1-mediated metabolic reprogramming promotes leukemogenesis

**Hairui Su[1], Yong Sun[2,3], Han Guo[4], Chiao-Wang Sun[1], Qiuying Chen[5], Szumam Liu[6], Anlun Li[6], Min Gao[7], Rui Zhao[1], Glen Raffel[8], Jian Jin[9], Cheng-Kui Qu[10], Michael Yu[11], Christopher A Klug[12], George Y Zheng[13], Scott Ballinger[2], Matthew Kutny[14], Long X Zheng[6], Zechen Chong[7], Chamara Senevirathne[4], Steven Gross[5], Yabing Chen[2,3]\*, Minkui Luo[4,5]\*, Xinyang Zhao[6]\***

[1]Department of Biochemistry and Molecular Genetics, The University of Alabama at Birmingham, School of Medicine, Birmingham, United States; [2]Department of Pathology, The University of Alabama at Birmingham, School of Medicine, Birmingham, United States; [3]Department of Pathology and Laboratory Medicine, Oregon Health and Science University and Research Department, Portland Veterans Affairs Medical Center, Portland, United States; [4]Tri-Institutional PhD Program of Chemical Biology, Chemical Biology Program, Memorial Sloan Kettering Cancer Center, New York, United States; [5]Program of Pharmacology, Weill Cornell Medical College of Cornell University, New York, United States; [6]Department of Pathology & Laboratory Medicine, University of Kansas Medical Center, Kansas City, United States; [7]Department of Genetics, The University of Alabama at Birmingham, School of Medicine, Birmingham, United States; [8]Department of Medicine, University of Massachusetts, Amherst Center, United States; [9]Center for Therapeutics Discovery, Mountain Sinai Hospital, Baltimore, United States; [10]Department of Pediatrics, Aflac Cancer and Blood Disorders Center, Children's Healthcare of Atlanta, School of Medicine, Emory University, Atlanta, United States; [11]Department of Biological Sciences, SUNY at Buffalo, Buffalo, United States; [12]Department of Microbiology, The University of Alabama at Birmingham, School of Medicine, Birmingham, United States; [13]Department of Pharmaceutical and Biomedical Sciences, College of Pharmacy University of Georgia, Athens, United States; [14]Department of Pediatrics, The University of Alabama at Birmingham, School of Medicine, Birmingham, United States

**\*For correspondence:**
chenyab@ohsu.edu (YC);
luom@mskcc.org (ML);
xzhao3@kumc.edu (XZ)

**Competing interest:** The authors declare that no competing interests exist.

## eLife Assessment

This study reveals that PRMT1 overexpression drives tumorigenesis of acute megakaryocytic leukemia (AMKL) and that targeting PRMT1 is a viable approach for treating AMKL. After revision, both reviewers found that these findings are **important** and that the data supporting these findings are **convincing**. Furthermore, these findings likely have significant implications for the treatment of AMKL with PRMT1 overexpression in the future.

**Abstract** Copious expression of protein arginine methyltransferase 1 (PRMT1) is associated with poor survival in many types of cancers, including acute myeloid leukemia. We observed that a specific acute megakaryocytic leukemia (AMKL) cell line (6133) derived from RBM15-MKL1 knock-in

mice exhibited heterogeneity in Prmt1 expression levels. Interestingly, only a subpopulation of 6133 cells expressing high levels of Prmt1 caused leukemia when transplanted into congenic mice. The PRMT1 inhibitor, MS023, effectively cured this PRMT1-driven leukemia. Seahorse analysis revealed that PRMT1 increased the extracellular acidification rate and decreased the oxygen consumption rate. Consistently, PRMT1 accelerated glucose consumption and led to the accumulation of lactic acid in the leukemia cells. The metabolomic analysis supported that PRMT1 stimulated the intracellular accumulation of lipids, which was further validated by fluorescence-activated cell sorting analysis with BODIPY 493/503. In line with fatty acid accumulation, PRMT1 downregulated the protein level of CPT1A, which is involved in the rate-limiting step of fatty acid oxidation. Furthermore, administering the glucose analog 2-deoxy-D-glucose delayed AMKL progression and promoted cell differentiation. Ectopic expression of Cpt1a rescued the proliferation of 6133 cells ectopically expressing PRMT1 in the glucose-minus medium. In conclusion, PRMT1 upregulates glycolysis and downregulates fatty acid oxidation to enhance the proliferation capability of AMKL cells.

## Introduction

Normal hematopoietic stem/progenitor cell transformation to acute myelogenous leukemia (AML) cells requires metabolic reprogramming (*Kreitz et al., 2019*). AML cells rely heavily on glucose for unchecked proliferation. Using $^{18}$F-Fluoro-deoxy-Glucose ($^{18}$FDG) as a marker, Cunningham et al. detected high glucose uptake in the bone marrow of AML patients, and pyruvate and 2-hydroxy-glutarate concentrations negatively correlate with patient survival rates (*Chen et al., 2014*). Dysregulated metabolic enzyme and mitochondrial activities have been reported to be the causes of chemoresistance to AML (*Lagadinou et al., 2013*; *Jones et al., 2019*) as well as solid tumors (*Faubert et al., 2020*). Metabolites such as acetyl-CoA, α-ketoglutarate, vitamin C (aka ascorbic acid), and *S*-adenosyl-L-methionine (SAM) are cofactors for histone and DNA modifications. Thus, metabolic reprogramming transforms the epigenetic landscape in leukemia cells. Mutations in isocitrate dehydrogenases, IDH1/2, produce 2-hydroxyl-glutarate (2-HG) instead of α-ketoglutarate. 2-HG inhibits demethylases that erase methylation marks on histones and DNA and hydroxylases such as FIH (factor inhibiting HIF) in leukemia and glioblastoma (*M. Gagné et al., 2017*). However, mutations of metabolic enzymes in cancer are relatively rare. Usually, metabolic reprogramming is achieved mainly by expressing oncogenic transcription factors such as p53 mutants, HIF1, and FOXOs (*Schwartzenberg-Bar-Yoseph et al., 2004*; *Choi et al., 2012*; *Laplante and Sabatini, 2013*; *Humpton et al., 2018*). Dysregulation of signaling pathways, such as KRAS mutations (*Ferrer et al., 2014*; *Faubert et al., 2020*) and upregulation of the mTOR pathway during leukemogenesis (*Kalaitzidis et al., 2012*), also alter metabolic pathways in tumorigenesis. Nevertheless, how epigenetic regulators involved in leukemogenesis regulate metabolic reprogramming still needs more research.

The protein arginine methyltransferase (PRMT) family has nine members, with PRMT1 responsible for most of the enzymatic activity in mammalian cells. PRMT1 is an epigenetic regulator via methylation of histone H4 and transcription factor RUNX1 (*Wang et al., 2001*; *Zhao et al., 2008*). The oncogenic roles of PRMT1 have been demonstrated in many types of solid cancers (*Le Romancer et al., 2008*; *Mitchell et al., 2009*; *Guendel et al., 2010*; *Karkhanis et al., 2011*; *Yoshimatsu et al., 2011*; *Baldwin et al., 2012*; *Cho et al., 2012b*; *Cho et al., 2012a*; *Yang and Bedford, 2013*; *Avasarala et al., 2015*). The importance of PRMT1 in leukemia has been shown in FLT3-ITD, AML1-ETO, and MLL-EEN-associated acute myeloid leukemia and lymphoid leukemia (*Lin et al., 1996*; *Cheung et al., 2007*; *Shia et al., 2012*; *Zou et al., 2012*; *He et al., 2019*). Targeting PRMT1 is effective in treating leukemia with splicing factor mutations (*Fong et al., 2019*). PRMT1 expression levels are low in quiescent hematopoietic stem cells but are elevated in stressed HSCs. Furthermore, the upregulation of PRMT1 enhances glycolysis through the methylation of PFKFB3 (*Watanuki et al., 2024*). Yet, how PRMT1 is involved in cancer metabolic reprogramming has not been explored, albeit the known role of PRMT1 in metabolic regulation in model organisms. Phosphorylation of Hmt1 (PRMT1 ortholog in yeast) controls cell cycle progression in response to nutrition signals (*Butcher et al., 2006*; *Messier et al., 2013*). PRMT1 in *Caenorhabditis elegans* and Trypanosoma is responsible for methylation of proteins inside mitochondria, although PRMT1 is not inside mitochondria, while PRMT1-null worms have dysfunctional mitochondria (*Fisk and Read, 2011*; *Sha et al., 2017*). In trypanosomes, PRMT1

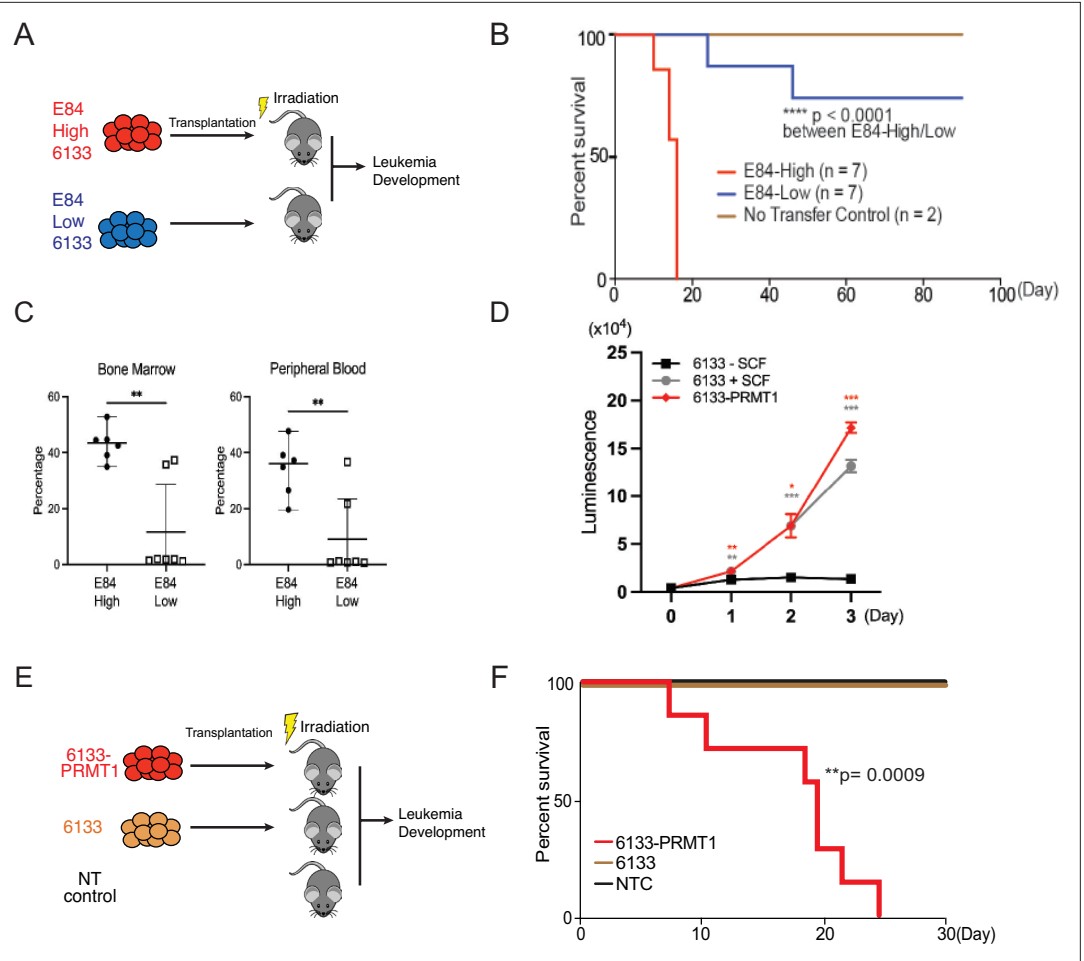

**Figure 1.** PRMT1 promotes leukemia cell transformation. (**A**) 6133cells were stained with E84, then fluorescence-activated cell sorting (FACS) was sorted based on E84 intensity. $3 \times 10^5$ sorted cells were intravenously transferred to sub-lethally irradiated mice. (**B**) Leukemia progression in recipient mice was shown on Kaplan–Meier curves. (**C**) The manifestation of leukemia cells in the bone marrow and peripheral blood of recipient mice was measured using flow cytometry. In the E84-low group, the bottom five dots represent five recipient mice that were sacrificed on day 90 post-transfer. Closed symbols indicate moribund mice, while open symbols denote non-terminally ill, inhibitor-treated mice sacrificed on day 88. (**D**) PRMT1 expression renders 6133cells' cytokine-independent growth. 6133cells and 6133/PRMT1 cells were cultured with or without mouse stem cell factor (SCF). Cell viabilities were measured daily. (**E**) Schematic for inducing leukemia through the intravenous injection of 6133 or 6133/PRMT1 cells into sub-lethally irradiated recipient mice ($n = 7$). (**F**) Leukemia progression in recipient mice was documented on Kaplan–Meier curves. * $p<0.05$, ** $p<0.01$, *** $p<0.001$.

The online version of this article includes the following figure supplement(s) for figure 1:

**Figure supplement 1.** Expression of metabolic regulated genes in cells with differential PRMT1 expression levels.

promotes glycolysis and is required for virulent infection (*Kafková et al., 2018*). When quiescent yeast re-entered fresh glucose-rich medium, Hmt1, the yeast PRMT1 homolog, is among nearly 240 genes that were induced (by ≥5-fold) in the presence of fermentable sugars like glucose, suggesting its crucial role in supporting cells under rapid growth and fermentation conditions (*Brejning et al., 2003*). When transitioning from glucose to non-fermentable carbon sources such as glycerol, Hmt1 is downregulated, implying that it is likely repressed to facilitate this vital adaptation (*Roberts and Hudson, 2006*).

Acute megakaryoblastic leukemia (AMKL) is a subtype of AML with leukemia cells stuck at the differentiation stage of immature megakaryocytes. It is a rare leukemia often associated with Down syndrome. In cases not related to Down syndrome, AMKL is caused by chromosomal translocations. Although AMKL can occur in adults, it occurs more commonly in children (*Athale et al., 2001*).

Chromosomal translocation t(1;22) that generates the RBM15-MKL1 fusion protein was discovered in childhood AMKL (*Ma et al., 2001*; *Mercher et al., 2001*). RBM15-MKL1 is a fatal disease without available targeted therapy.

Copious expression of PRMT1 is a poor prognostic marker for AML (*Zhang et al., 2015*; *Zhu et al., 2019*). Furthermore, PRMT1 is expressed at an even higher level in AMKL than in other types of AML (*Zhang et al., 2015*). Constitutive expression of PRMT1 blocks terminal MK differentiation (*Zhang et al., 2015*), while inhibition of PRMT1 activity promotes terminal MK differentiation (*Su et al., 2021*). Thus, we hypothesize that inhibiting PRMT1 activity could be a pro-differentiation therapy for AMKL. A leukemia cell line called 6133 is derived from Rbm15-MKL1 knock-in mice. When transplanted, 6133 cells can cause AMKL with low penetrance (*Mercher et al., 2009*). Using this leukemia mouse model, we report here that the elevated level of PRMT1 maintains the leukemic cells via upregulation of glycolysis and that leukemia cells with high PRMT1 expression are vulnerable to the inhibition of fatty acid metabolic pathways.

## Results

### PRMT1 promotes the progression of RBM15-MKL1-initiated leukemia

The 6133 cells can be transplanted into recipient mice to induce low penetrant leukemia with symptoms closely recapitulating human AMKL (*Mercher et al., 2009*). To find additional factors needed to transform 6133 cells fully, we have reported a fluorescent probe (E84) that can be used to sort live cells according to PRMT1 protein concentrations (*Su et al., 2018*; *Figure 1—figure supplement 1*). We sorted 6133 cells into two populations for bone marrow transplantation (BMT) according to E84 staining intensities (*Figure 1A*). All mice that received 6133 cells expressing higher levels of PRMT1 (6133/PRMT1 cells) developed leukemia and died rapidly, while 6133 cells expressing lower levels of PRMT1 did not develop leukemia (*Figure 1B*). Consistently, a higher percentage of leukemia cells were detected in bone marrow and peripheral blood in recipient mice transplanted with E84-high (i.e., density staining of E84) 6133 cells according to fluorescence-activated cell sorting (FACS) analysis (*Figure 1C*).

Given that E84-high cells can initiate leukemia, we introduced PRMT1 into 6133 cells (aka 6133/PRMT1 cells) using a lentivirus vector. Overexpression of PRMT1 rendered 6133/PRMT1 cells to increase in a cytokine-independent fashion in cell culture (*Figure 1D*), and recipient mice transplanted with 6133/PRMT1 cells developed leukemia and died within 25 days (*Figure 1E, F*). Although PRMT1-mediated methylation triggers the degradation of RBM15, PRMT1 overexpression does not affect the stability of the RBM15-MKL1 fusion (*Figure 2—figure supplement 1A and B*). The leukemic mice displayed splenomegaly. Intriguingly, the leukemia mice were paralyzed with observable spinal bleeding during dissection, although the bone density was still normal (data not shown). In a xenograft model with human RBM15-MKL1 leukemia cells, leukemia also caused spinal bleeding (*Thiollier et al., 2012*). Collectively, PRMT1 is essential for the initiation of overt AMKL expressing the RBM15-MKL1 fusion protein, and leukemia initiated by 6133/PRMT1 cells in mice resembles several characteristics of human AMKL.

### MS023, a PRMT1 inhibitor, cures mice with Rbm15-MKL1-initiated leukemia

MS023 was reported to be a potent inhibitor of Type-I PRMTs including PRMT1 (*Eram et al., 2016*; *Su et al., 2021*). Furthermore, MS023 has been tested safe on mice at 80 mg/kg of body weight (*He et al., 2019*; *Su et al., 2021*). The 6133/PRMT1 cells were intravenously injected into sub-lethally irradiated recipient mice. A week after BMT, we injected MS023 intraperitoneally every other day for a month. Notably, while the untreated group of mice exhibited rapid illness and developed moribund symptoms, such as severe weight loss and rear limb paralysis, within 30 days, the majority of mice treated with MS023 remained healthy and symptom-free. Treatment with MS023 significantly alleviated the leukemia burden, as shown by FACS analysis of the percentages of leukemia cells in bone marrow and peripheral blood (*Figure 2D*). We also validated that the bone marrow cells from MS023-treated mice had reduced global levels of arginine methylation (*Figure 2—figure supplement 1C*). Leukemia-associated splenomegaly was also alleviated in MS023-treated mice (*Figure 2D*). Kaplan–Meier curves showed that the MS023-treated group was effectively cured after 120 days (*Figure 2C*), with only

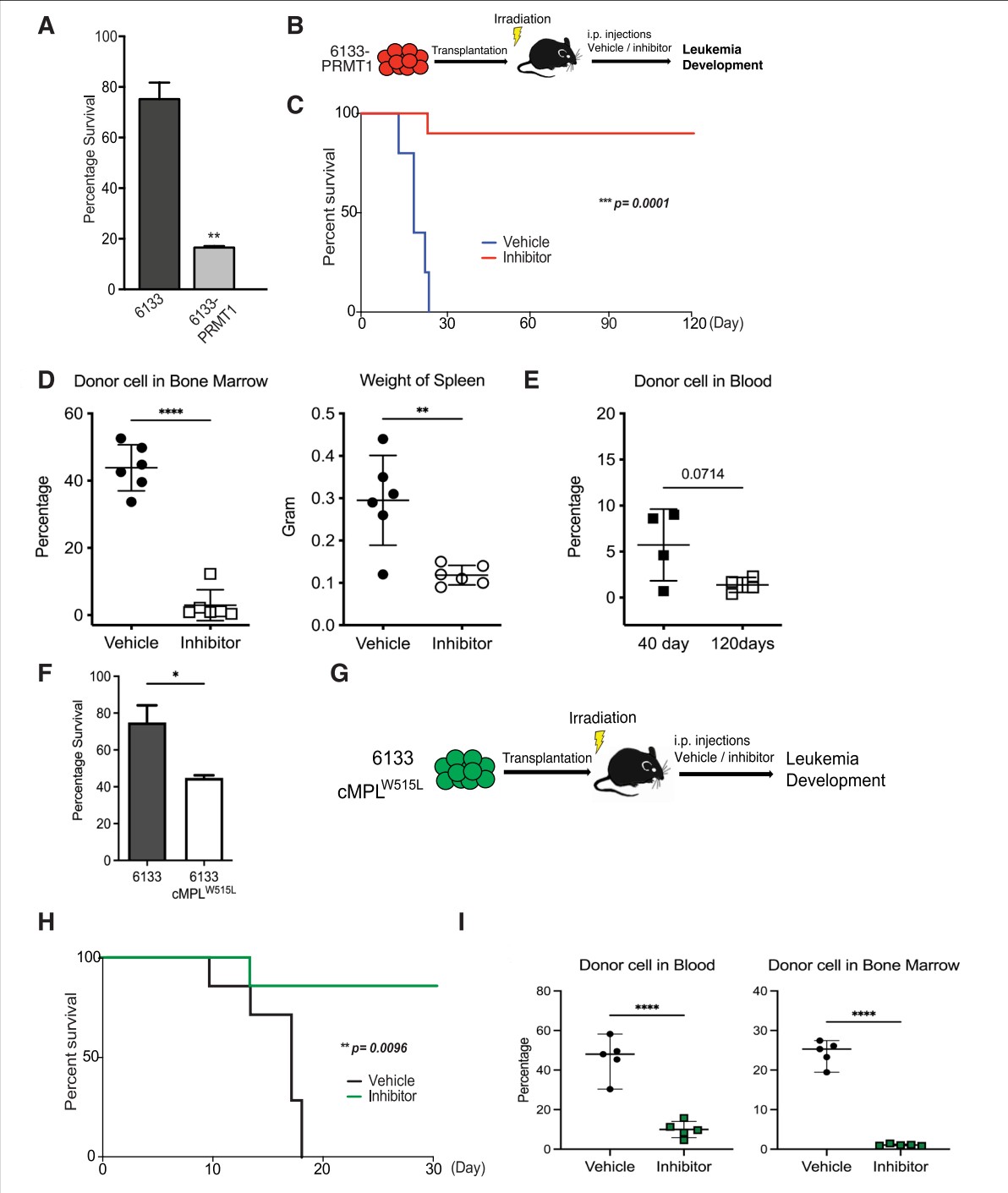

**Figure 2.** PRMT1 inhibitor MS023 blocks leukemia progression. (**A**) The survival of 6133/PRMT1 cells is highly sensitive to treatment with MS023. Both 6133 and 6133/PRMT1 cells were treated with MS023 for 48hr, and cell viability was determined by counting. (**B**) 6133/PRMT1 cells were intravenously transferred into sub-lethally irradiated mice. Recipient mice received intraperitoneal injections of either a PRMT1 inhibitor or vehicle for 15 doses, given every other day. (**C**) The progression of leukemia was illustrated on Kaplan–Meier curves. *n* = 5. (**D**) Leukemia cells in recipient mice (*n* = 6) were quantified using flow cytometry. The right panel shows the weights of the spleens from recipient mice. Closed symbols represent moribund mice, and open symbols represent non-terminally ill, inhibitor-treated mice that were sacrificed on days 40 and 120. (**E**) Peripheral blood was collected from non-terminally ill, inhibitor-treated mice at 40 and 120days post-cell transfer. (**F**) MS203 treatment of in vitro cultured 6133 and 6133/cMPLW515L cells. *n* = 3. p < 0.05. (**G**) Schematic of leukemia induced by 6133 c-mplW515L transplantation. (**H**) Kaplan–Meier curves for MS023-treated leukemia mice induced by 6133 c-mplW515L cells. *n* = 7. (**I**) The percentage of GFP-positive leukemia cells in the peripheral blood and bone marrow of the vector and inhibitor-treated mice at the endpoints. * p<0.05, ** p<0.01, **** p<0.0001.

*Figure 2 continued on next page*

*Figure 2 continued*

The online version of this article includes the following source data and figure supplement(s) for figure 2:

**Figure supplement 1—source data 1.** Labeled gel for the western blots in *Figure 2—figure supplement 1*.

**Figure supplement 1—source data 2.** Western blots for the *Figure 2—figure supplement 1*.

**Figure supplement 1—source data 3.** Western blots with labels for *Figure 2—figure supplement 1B*.

**Figure supplement 1—source data 4.** *Figure 2—figure supplement 1* western blot raw data.

**Figure supplement 1.** RBM15-MKL1 (OTT-MAL) protein stability is not affected by PRMT1 activity.

residual leukemia cells detected in bone marrow and peripheral blood (*Figure 2D, E*). Expression of the c-MPLW515L mutant in 6133 cells can render the 6133 cells fully penetrant for leukemia (*Mercher et al., 2009*), and in vitro treatment with MS023 reduced their proliferation (*Figure 2F*). Subsequently, we transplanted 6133/cMPLW515L cells into congenic mice. *Figure 2G–I* illustrates that MS023 also cures leukemia, suggesting that PRMT1 could be a valid target for leukemia or myeloid proliferative diseases driven by c-MPLW515L. Collectively, these data demonstrate that PRMT1 is critical for sustaining Rbm15-MKL1-initiated leukemia in mice, and pharmacological targeting of PRMT1 represents an effective strategy for treating leukemia.

## PRMT1 promotes glycolysis in leukemia cells

Given that the role of PRMT1 in metabolic regulation has been well documented, we then investigated whether PRMT1 is involved in transforming 6133 cells through metabolic reprogramming. To assess this, we conducted Seahorse assays to measure the changes in ECAR (extracellular acidification rate) in 6133/PRMT1 cells compared to the parental 6133 cells. The ECAR curve of 6133/PRMT1 cells was elevated compared to that of 6133 cells (*Figure 3A*). Interestingly, the addition of Carbonyl cyanide 4-(trifluoromethoxy)phenylhydrazone (FCCP), which uncouples oxidative phosphorylation from the tricarboxylic acid (TCA) cycle, did not increase acidification levels. Similarly, antimycin did not cause a significant drop in acid concentration. Remarkably, we treated 6133 cells with MS023 overnight prior to Seahorse assays and observed a reduction in ECAR levels (*Figure 3B*). These findings suggest that PRMT1 is responsible for the observed increase in acidification. In this experiment, glycolysis contributes the most to acidification in the 6133 leukemia cells, as the addition of mitochondrial respiratory inhibitors only moderately reduces acidification levels according to the principles outlined (*Divakaruni et al., 2014*). Lactate dehydrogenase A (LDHA), the final key enzyme in glycolysis that converts pyruvate to lactate for $NAD^+$ regeneration, was found to be influenced by PRMT1 overexpression. Specifically, PRMT1 overexpression stimulated the tyrosine phosphorylation of LDHA, thereby activating its enzymatic activity (*Fan et al., 2011*) despite an overall decrease in LDHA levels (*Figure 3C*). Consistently, when we directly measured the intracellular and extracellular lactate levels in 6133 and 6133/PRMT1 cells, we observed that 6133/PRMT1 cells not only released more lactate into the medium but also exhibited higher intracellular lactate levels (*Figure 3D*). We concluded that PRMT1 promotes cellular glycolysis and lactate production, which predominantly contributes to the observed increase in acidification.

## PRMT1 reduces oxygen consumption in mitochondria

The Seahorse assays have demonstrated that the cellular OCR (oxygen consumption rate) was reduced in 6133/PRMT1 cells. Additionally, it was observed that these cells had a limited capacity for reserve respiration, as the maximum respiration level was nearly identical to the basal OCAR level, regardless of the expression levels of PRMT1 (*Figure 4A*). Conversely, when the 6133/PRMT1 cells were pre-treated with MS023 overnight before the Seahorse assays, there was an increase in mitochondrial oxygen consumption compared to the non-treated controls (*Figure 4B*). These findings further support the notion that PRMT1 plays a role in mediating metabolic reprogramming by enhancing glycolysis and reducing mitochondrial oxygen consumption.

We demonstrated that PRMT1 increases the number of mitochondria in MEG-01 cells, a human AMKL cell line, by confocal microscopy (*Zhang et al., 2015*). In this experiment, FACS analysis with MitoTracker staining (*Figure 4C*) and real-time PCR analysis of mitochondrial DNA (*Figure 4D*) further validated that PRMT1 upregulated mitochondrial biogenesis in 6133 cells. H2DCFDA staining showed that PRMT1 elevated global ROS levels in 6133 cells (*Figure 4F*). Interestingly, when we performed

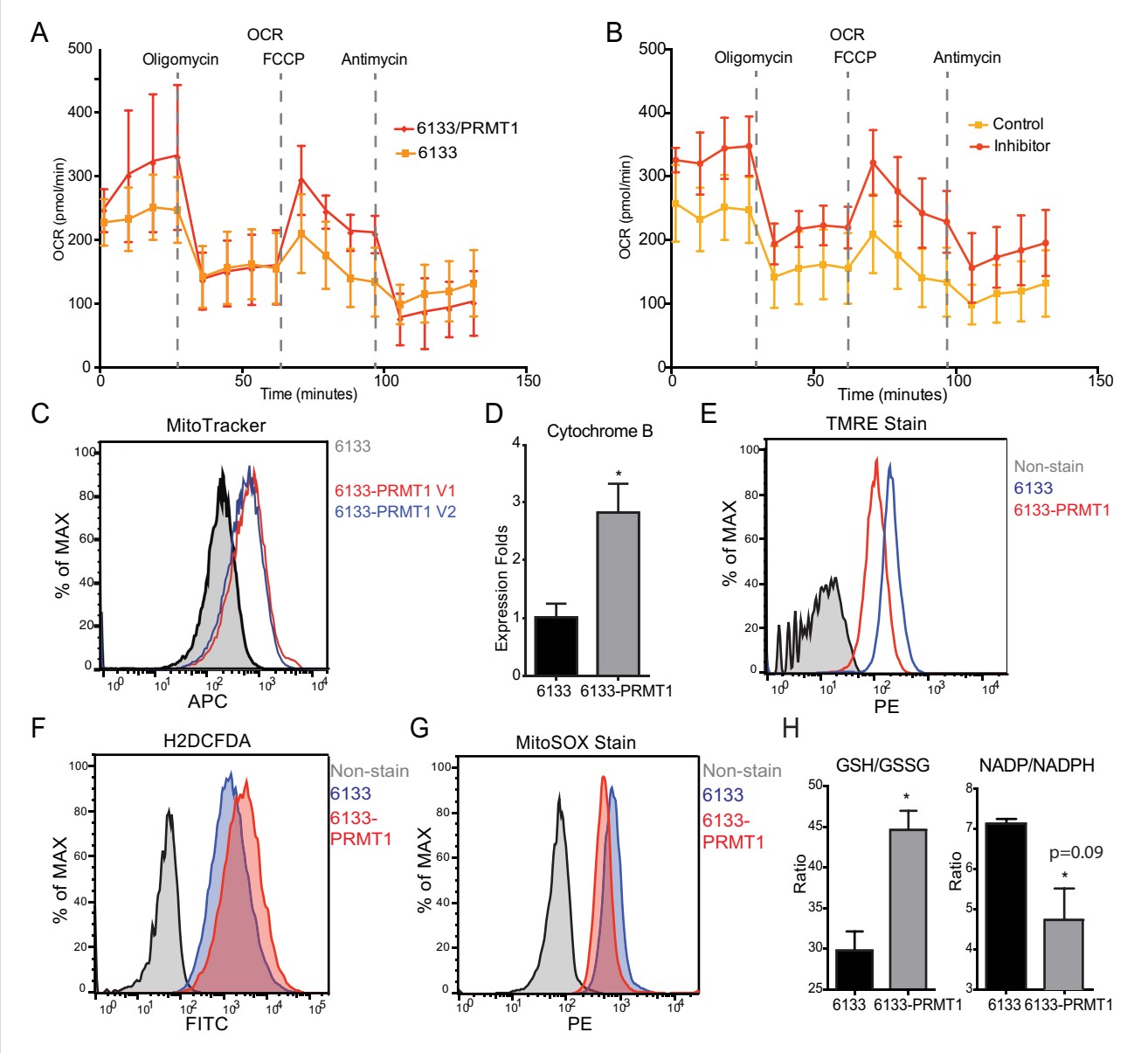

**Figure 3.** PRMT1 promotes glycolysis. (**A**) Extracellular acidification rates (ECARs) in 6133 and 6133/PRMT1 cells were measured using Seahorse assays. 150,000cells were seeded in special 24-well plates for the Seahorse Xf-24 analyzer. Oligomycin, FCCP, and antimycin were injected sequentially into the wells as indicated. (**B**) ECARs in PRMT1 inhibitor (MS023)-treated 6133cells. Cells were pretreated with MS023 overnight before Seahorse analysis. (**C**) The protein levels of lactate dehydrogenase A (LDHA) and p-Y10-LDHA in both 6133 and 6133/PRMT1 cells were assessed using western blotting. The cells were cultured at the same density, and extracts were harvested after an overnight incubation. (**D**) Intracellular and extracellular lactate levels were measured using an L-lactate kit. Cells were seeded at $1 \times 10^7$cells/ml and cultured for 24hr, followed by centrifugation. Both the medium/supernatant (extracellular) and the cell pellet (intracellular) were collected for analysis. The results from the triplicates are plotted. *$p < 0.05$.

The online version of this article includes the following source data for figure 3:

**Source data 1.** Pdf files containing original western blots for *Figure 3C*, indicating the relevant bands and treatments.

**Source data 2.** JPG files containing original western blots for *Figure 3D*.

Mito-Sox staining for ROS levels within mitochondria, we found reduced levels of ROS (*Figure 4G*). These data imply that most of the ROS detected by H2DCFA may be generated from the cytoplasm rather than from mitochondria, which aligns with the Seahorse data indicating that oxidation consumption in mitochondria is reduced (*Figure 4A*). Next, we conducted TMRE staining for mitochondrial membrane potential. 6133/PRMT1 cells exhibited reduced staining, suggesting that PRMT1

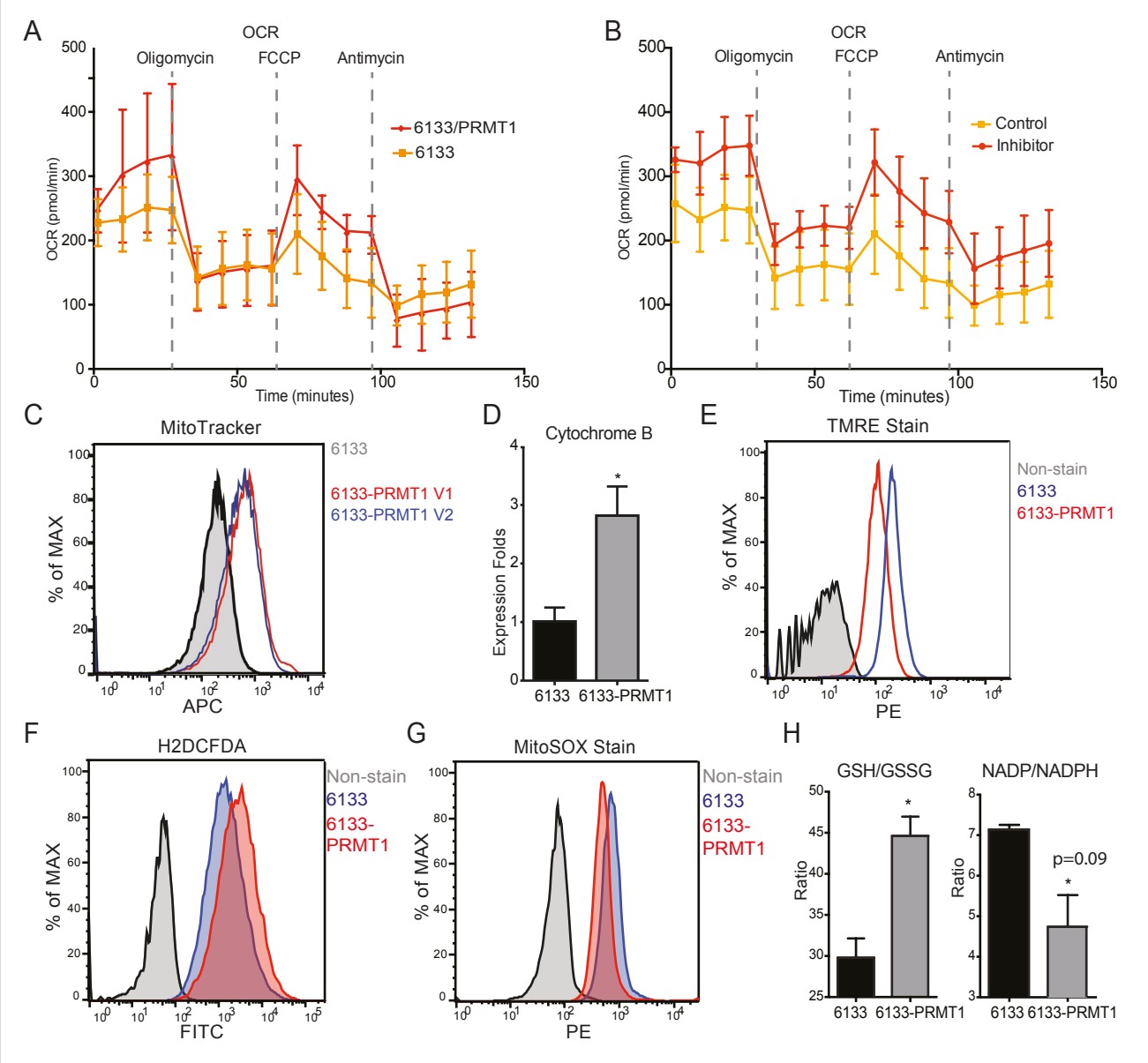

**Figure 4.** PRMT1 changes oxygen consumption and redox status. (**A**) Oxygen consumption rates (OCRs) in 6133 and 6133/PRMT1 cells were measured using Seahorse assays. A total of 150,000cells were seeded in specialized 24-well plates designed for the Seahorse Xf-24 analyzer. Oligomycin, FCCP, and antimycin were injected into the wells sequentially as indicated. (**B**) OCRs in PRMT1 inhibitor-treated 6133/PRMT1 cells. Cells were pretreated with PRMT1 inhibitor MS023 overnight prior to Seahorse analysis. (**C**) Mitotracker Deep Red FM staining assessed the mitochondrial mass in 6133 and 6133/PRMT1 cells. (**D**) Quantitative PCR of mitochondria-specific gene cytochrome B measured the mitochondrial DNA amount. (**E**) TMRE staining was performed to measure mitochondrial membrane potential. Cells were seeded at $1.1 \times 10^5$cells/ml, and 0.1volume of 5mM TMRE was added to the culture to reach a final concentration of 500nM. After 20min, cells were collected and used for fluorescence-activated cell sorting (FACS) analysis. (**F**) The intracellular ROS level was measured by H2-DCFDA staining. $1 \times 10^5$cells were incubated in a warm staining solution containing 10μM of H2DCFDA for 30min, then washed and subjected to analysis. (**G**) Mitochondrial ROS was measured by MitoSOX staining. $5 \times 10^5$cells were incubated with a warm staining solution containing 2.5μM of MitoSOX for 10min and then washed and subjected to analysis. (**H**) Intracellular levels of GSH/GSSG and NADP/NADPH ratio were measured. In each assay, $5 \times 10^5$cells were used for extract preparation. * $p<0.05$.

decreases membrane potential (*Figure 4E*). This result also cross-validated the Seahorse findings that PRMT1 reduced mitochondrial oxygen consumption.

Since the cytoplasmic ROS level significantly increased with PRMT1 expression, we measured the ratios of two redox pairs: glutathione and NADP+. Interestingly, activation of PRMT1 raised the ratios of both GSH/GSSG and NADPH/NADP+. Given the crucial roles of GSH and NADPH in biomass

synthesis, the upregulation of PRMT1 may promote anabolism, consistent with PRMT1's role in supporting cell proliferation.

## Metabolomic analysis of PRMT1-induced metabolic changes

To further probe the complexity of PRMT1-mediated metabolic reprogramming, we established a new 6133 cell line that can conditionally express PRMT1 upon adding doxycycline to the medium. We compared the metabolomic status of 6133 cells before and after 12 hr of PRMT1 induction. The cells were grown in standard RPMI 1640 medium with 10% fetal bovine serum, with high concentrations of glucose and essential amino acids for 12-hr growth. We repeated the metabolomic analysis five times. Principal component analysis indicated that PRMT1 caused a significant shift in metabolite profiles (*Figure 5A, B*, *Figure 5—source data 1*). Notably, PRMT1 activation led to increased ATP production and the accumulation of succinyl-CoA, alanine, serine, and short-chain fatty acids and depletion of SAM, aspartic acid, nicotinamide, succinyl-homoserine, oxidized glutathione (which is consistent with the increased ratio of GSH/GSSG shown in *Figure 4H*), and alpha-ketoglutarate (aka oxoglutaric acid) (*Figure 5C*). These findings suggest that the beta-oxidation of fatty acids may be impaired, leading to the accumulation of fatty acids such as docosadienoic acid, caproic acid, and suberic acid, while the levels of palmitoyl-carnitine are reduced. Furthermore, the reduced levels of alpha-ketoglutarate indicate potential alterations in the TCA cycle and glutaminolysis. Overall, our data highlight the profound metabolic changes induced by PRMT1, particularly in amino acid biosynthesis and one-carbon metabolism, as shown in *Figure 5D* generated by the metaboAnalyst software (*Pang et al., 2024*).

## PRMT1-transformed leukemia cells are highly dependent on glucose consumption

After determining that glycolysis was enhanced in PRMT1-overexpressing cells, we performed glucose colorimetric assays to measure glucose concentrations in the cell culture medium when the 6133 and 6133/PRMT1 cells grew exponentially. A more significant reduction of glucose in 6133/PRMT1 cells was observed compared to parental 6133 cells (*Figure 6A*), indicating that the elevated glucose consumption via glycolysis is due to PRMT1 upregulation. Next, we cultured the cells in both glucose-containing and glucose-depleted media. Cell viability assays showed that 6133/PRMT1 cells grew more slowly in a glucose-free medium, while parental cells were less affected (*Figure 6B*). Accordingly, we used 2-deoxy-D-glucose (2-DG), a glucose analog that competes with glucose in glycolysis. The proliferation of 6133/PRMT1 cells was notably inhibited by 2-DG, while the proliferation of 6133 cells was barely affected (*Figure 6C*). These results are consistent with Seahorse analysis indicating that 6133/PRMT1 cells rely on glycolysis for growth (*Figure 3*). Consistently, we also demonstrated that E84-high 6133 cells will die quicker than the E84-low 6133 cells in glucose-minus medium (*Figure 6—figure supplement 1*). Next, we tested whether blocking glucose could impede AMKL progression in mice. Four days after the transplantation of the 6133/PRMT1 cells, we administered 2-DG to the mice at a dosage of 0.5 g/kg body weight every other day for a month. 2-DG treatment significantly delayed disease progression (*Figure 6D*). Splenomegaly was alleviated, and the burden of leukemia cells in the bone marrow and peripheral blood was notably decreased in the 2-DG treated mice (*Figure 6E, F*). Taken together, the in vitro and in vivo data suggest that PRMT1-mediated metabolic reprogramming renders leukemia cells highly dependent on glucose (*Figure 6—figure supplement 2*).

## Targeting fatty acid metabolic pathways for leukemia therapy

Rapidly growing cancer cells need de novo synthesis of fatty acids to meet the demand for building cell membranes (*Menendez and Lupu, 2007*). De novo fatty acid synthesis uses acetyl-CoA generated from fatty acid oxidation. Inhibition of fatty acid oxidation has been shown to kill leukemia cells (*Samudio et al., 2010*). Carnitine-palmitoyl-transferase (CPT1A) catalyzes the rate-limiting step of fatty acid oxidation by transporting fatty acids across the mitochondrial outer membrane. RBM15 binds to the 3' UTR of CPT1A mRNA (*Zhang et al., 2015*). Furthermore, PRMT1 regulates the stability of the RBM15 protein. We demonstrated that the expression of both isoforms of PRMT1 lowered the mRNA levels of CPT1A (*Figure 7A*). We then performed western blotting and confirmed that the protein level of CPT1A was correspondingly reduced in 6133 cell lines ectopically expressing PRMT1 isoforms (*Figure 6B*). The downregulation of CPT1A leads to reduced consumption of long-chain fatty acids. We consistently observed greater lipid accumulation in 6133/PRMT1 cells than in 6133 cells through

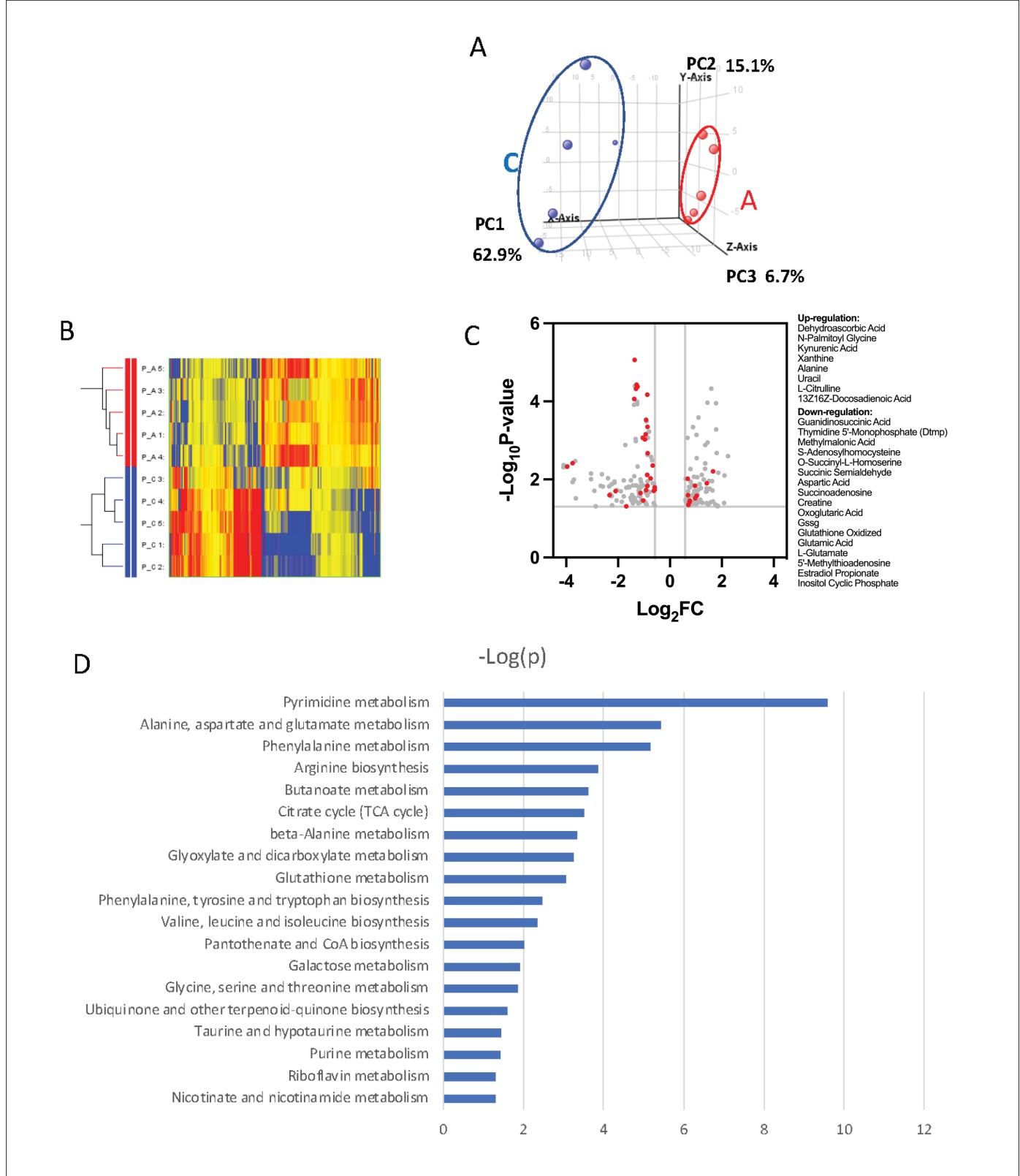

**Figure 5.** Metabolomic analysis of PRMT1-regulated metabolism in 6133cells. 6133cells expressing doxycycline-inducible PRMT1 were induced to express PRMT1. The activated group (labeled as **A**): metabolites collected after PRMT1 was induced. Control group (labeled as **C**): metabolites collected before PRMT1 was induced. (**A**) Principal component analysis (PCA) for the metabolites. The C and A groups were clustered at different positions in PCA analysis. (**B**) Heatmap of metabolites differentially expressed in these two groups. (**C**) Volcano plots for metabolites. 204 metabolites are changed more

*Figure 5 continued on next page*

*Figure 5 continued*
than twofolds with p < 0.5. (**D**) Metabolic pathways are primarily influenced by PRMT1 overexpression in 6133cells. The metabolite data were analyzed using a web-based program called MetaboAnalystR6.0.

The online version of this article includes the following source data for figure 5:

**Source data 1.** The metabolomic data for the PRMT1 mediated changes in metabolites.

FACS analysis with a lipid-binding fluorescent dye, that is BODIPY 493/503 (*Figure 7C*). Metabolomic analysis also indicated the accumulation of fatty acids such as suberic acid and caproic acid, alongside a decrease in palmitoyl-carnitine, which is a product of CPT1A (*Figure 5—source data 1* for metabolomic data). We then treated leukemia cells with etomoxir, which inhibits CPT1A enzymatic activity. The 6133/PRMT1 cells were more sensitive to etomoxir than the 6133 cells (*Figure 6D*). Orlistat, a FASN inhibitor, also suppressed the proliferation of 6133/PRMT1 cells more effectively than that of 6133 cells (*Figure 6E*). While CPT1A is responsible for transporting long-chain fatty acids, short-chain fatty acids can be directly transported to mitochondria. We added acetate, propionate, and butyrate in the form of triglycerides to a glucose-free medium. Strikingly, all three forms of short-chain fatty acids supported the proliferation of 6133/PRMT1 cells better than the parental 6133 cells (*Figure 6F*). The data suggest that leukemia cells with elevated levels of PRMT1 expression can utilize short-chain fatty acids to compensate for their need for glucose.

We then used a lentivirus to express CPT1A in 6133/PRMT1 cells ectopically. CPT1A wild type and mutant (H473A) that does not have succinylation activity (*Kurmi et al., 2018*) dampens the proliferation rates of 6133/PRMT1 cells cultured under standard conditions. Intriguingly, the enzymatically inactive CPT1A mutant (G710E) inhibits the proliferation of the 6133/PRMT1 cells, suggesting that the 6133 cells still rely on fatty acid oxidation for de novo lipid synthesis necessary for proliferation (*Figure 7G*). Taken together, we demonstrated that although PRMT1 downregulates fatty acid oxidation as an energy source, PRMT1 upregulates using short-chain fatty acids as an alternative energy source for mitochondria.

## Discussion

This report demonstrates that PRMT1 promotes glycolysis and reprograms mitochondrial metabolism to support leukemia progression in mice with RBM15-MKL1-initiated leukemia. Notably, only leukemia cells with elevated levels of PRMT1 can be transplanted to initiate leukemia (*Figure 1*). This indicates that PRMT1 assists leukemia cells in adapting to the bone marrow niche necessary for leukemia initiation, suggesting that leukemia stem cells likely express high levels of PRMT1. Moreover, since normal hematopoietic stem cells express low levels of PRMT (*Su et al., 2018*), targeting this protein may help protect normal hematopoietic stem cells from damage. Considering the crucial role of PRMT1 in acute myeloid leukemia (AML) with various gene mutations—including RBM15-MKL1, FLT3-ITD, MLL-EEN, and AML1-ETO fusions—as well as mutations in splicing factor (*Fong et al., 2019*), PRMT1-mediated metabolic reprogramming may be significant for the previously mentioned AML.

The potential mechanism of PRMT1-mediated metabolic reprogramming can be learned from published data. PRMT1 methylates phosphoglycerate kinase 1 (PGK1) at arginine 206, which enhances its phosphorylation at serine 203 (*Liu et al., 2024*). Additionally, PRMT1 methylates phosphoglycerate dehydrogenase at arginine 236, activating the enzyme and stimulating the synthesis of serine (*Wang et al., 2023*; *Yamamoto et al., 2024*). Furthermore, PRMT1-mediated methylation of glyceraldehyde 3-phosphate dehydrogenase prevents its localization in the nucleus of LPS-activated macrophages, protecting these macrophages from apoptosis (*Cho et al., 2018*). Overall, these findings underscore PRMT1's significant role in boosting glycolytic activity and tumorigenesis through the targeted methylation of glycolytic enzymes.

RBM15 plays a crucial role in regulating the homeostasis of hematopoietic stem cells and the differentiation of megakaryocytes. Our previous paper in eLife demonstrated that RBM15 is methylated by PRMT1, and this modification affects RBM15-mediated RNA splicing of key transcription factors, including RUNX1, GATA1, TAL1, and TPOR (also known as c-mpl), which are essential for megakaryocyte development (*Zhang et al., 2015*). Notably, we found that nearly 50% of RBM15 targets are metabolic enzymes, with their mRNAs' 3′ UTR regions bound by RBM15. Among these targets, CPT1A mRNA stands out as particularly significant. Our metabolomic analysis indicated that overexpression

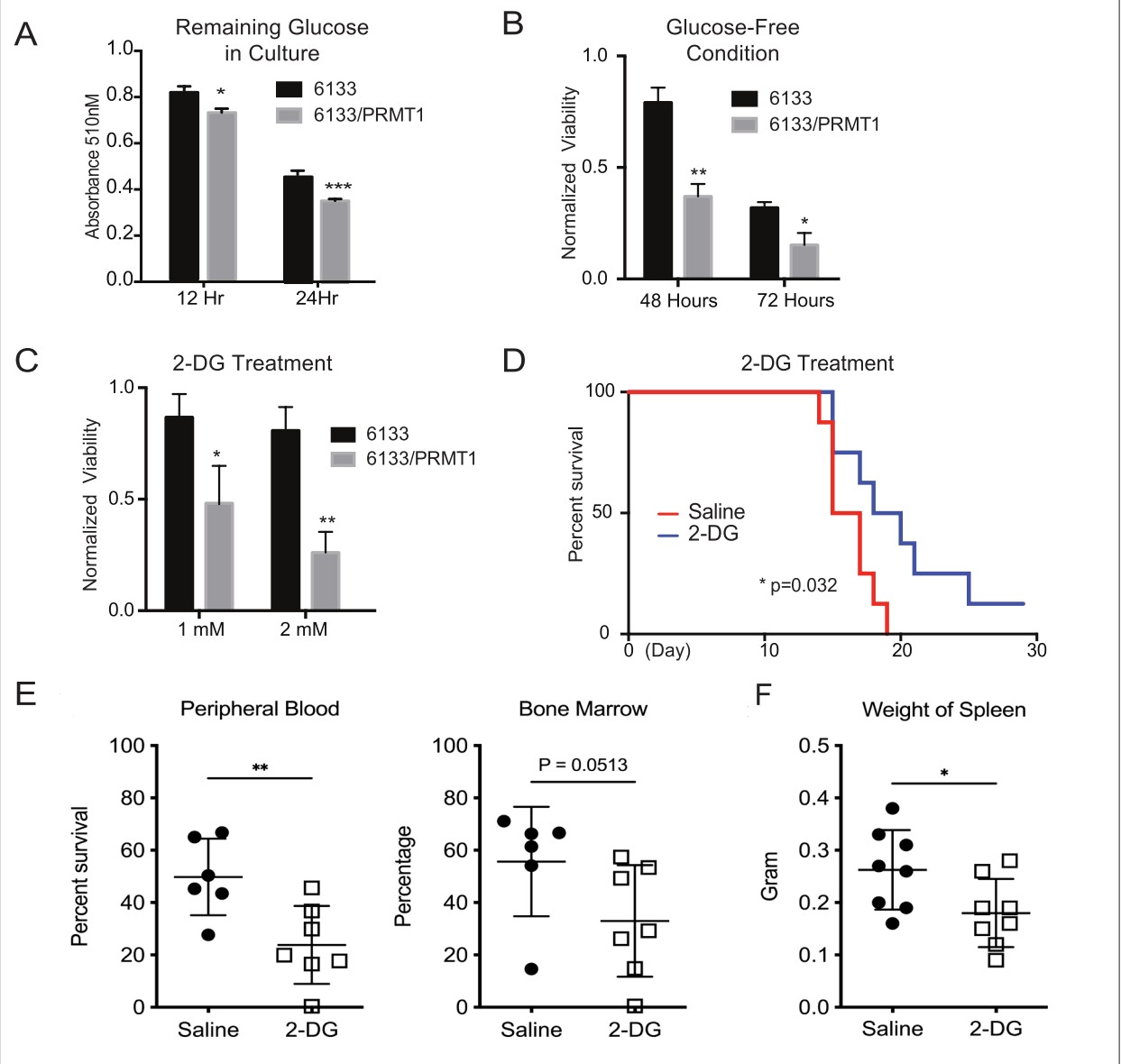

**Figure 6.** PRMT1 causes leukemia cells' heavy dependency on glucose consumption. (**A**) Colorimetric assay of glucose in 6133 and 6133/PRMT1 cells. $1.25 \times 10^5$ cells were seeded per well and cultured overnight. After centrifugation, the supernatant/medium was collected and used for the assay. (**B**) Cell viability of 6133 and 6133/PRMT1 cells under glucose-free conditions. Cells were seeded in a 96-well plate with fresh regular or glucose-minus RPMI 1640 medium. Cell viability was measured by CellTiter-Glo kit. Ratios were normalized to the wells with regular RPMI 1640 medium. (**C**) Cell viability of 6133 and 6133/PRMT1 cells following 2-deoxy-D-glucose (2-DG) treatment. Cells were plated in 96-well plates containing 2-DG. Cell viability was measured by CellTiter-Glo kit. The growth of cells without the addition of 2-DG served as a normalization reference. (**D**) 6133/PRMT1 cells were transferred intravenously into sub-lethally irradiated mice. Beginning on day 4 post-transfer, recipient mice received intraperitoneal injections of saline or 0.25g/kg body weight of 2-DG every other day. The survival of recipient mice is presented as Kaplan–Meier curves. (**E**) The percentages of leukemia cells in the bone marrow and peripheral blood of recipient mice were measured using flow cytometry. Closed symbols represent mice transplanted with 6133/PRMT1 cell-induced leukemia, while the open square indicates a control wild-type mouse without leukemia. * p<0.05, ** p<0.01, *** p<0.001.

The online version of this article includes the following figure supplement(s) for figure 6:

**Figure supplement 1.** Responsiveness to glucose restriction of cells with differential PRMT1 expression levels.

**Figure supplement 2.** Graphic abstract to summarize the PRMT1-mediated metabolic reprogramming in leukemia cells.

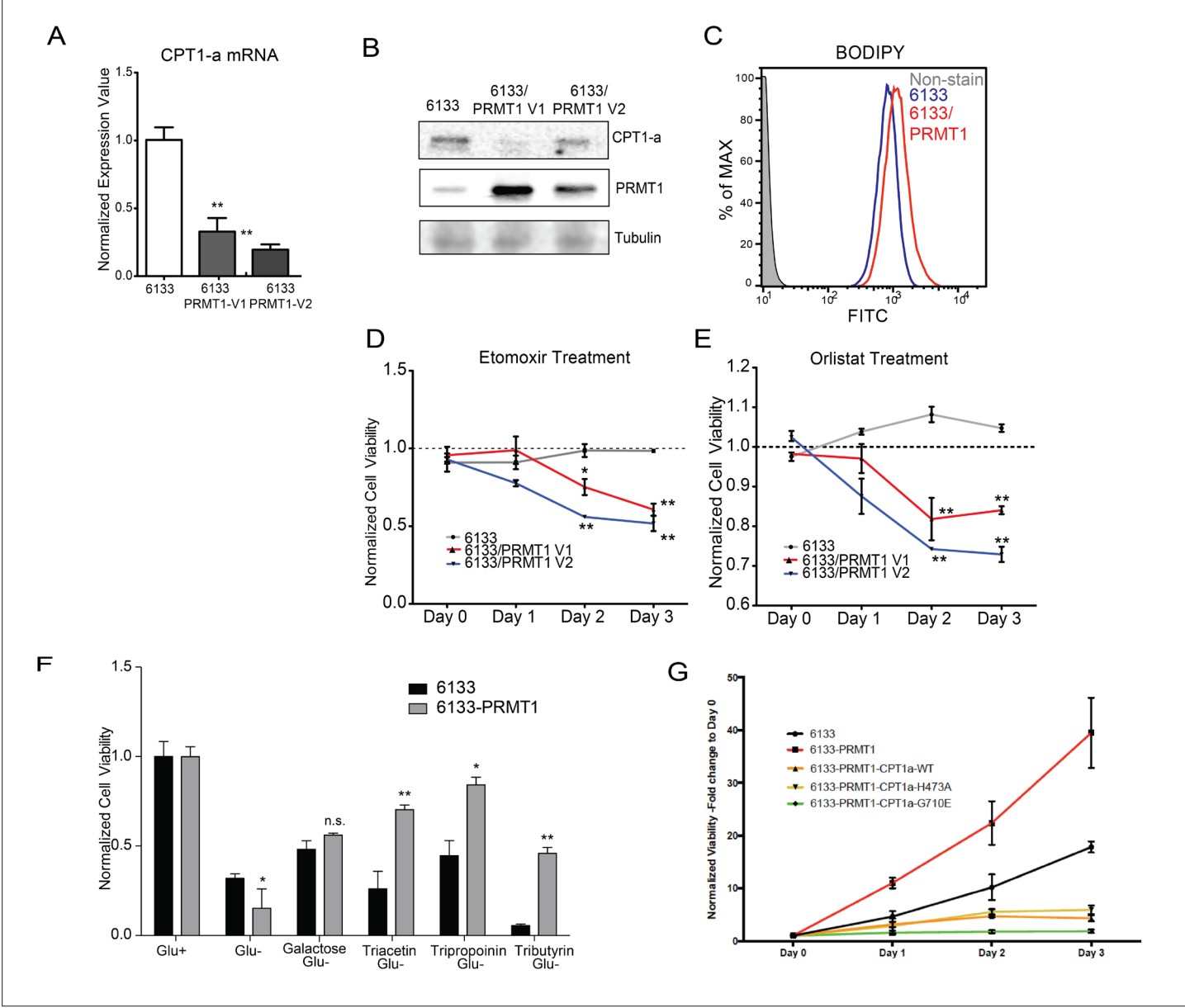

**Figure 7.** PRMT1 alters fatty acid oxidation in leukemia cells. (**A**) The mRNA levels of CPT1a and PPARα in 6133 and 6133/PRMT1 cells were assessed. Cell pellets were harvested in Trizol prior to RNA extraction, followed by cDNA synthesis and qPCR analysis. (**B**) Western blotting of 6133 and 6133/PRMT1 cell lines. (**C**) BODIPY/lipid Staining of 6133cells. Cells were incubated with 200nM of BODIPY/lipid at 37°C for 15min, then washed with medium and prepared for fluorescence-activated cell sorting (FACS) analysis. (**D**) Cell viability of 6133 and 6133/PRMT1 cells with Etomoxir treatment. 6133 and 6133-PRMT1 cells were seeded in a 96-well plate, supplemented with Etomoxir. Cell viability was measured by CellTiter-Glo Kit. Ratios were normalized to day 0. (**E**) Cell viability of 6133 and 6133-PRMT1 cells with Orlistat treatment. (**F**) Viability of 6133 and 6133-PRMT1 cells with a supplement of short-chain fatty acid (triacetin/tripropionin/tributyrin) under glucose-free conditions. Cells were cultured with RPMI with or without glucose, supplemented with 100nM of triacetin/tripropionin/tributyrin, respectively. Cell viability was measured after 72hr. Fold change of viability is normalized to glucose+ cells. (**G**) Growth curves of 6133-PRMT1 cells transduced with CPT1A, CPT1A/H473A, and CPT1A/G710E. * p<0.05, ** p<0.01.

The online version of this article includes the following source data for figure 7:

**Source data 1.** Pdf files containing original western blots for **Figure 7B**, indicating the relevant bands and treatments.

**Source data 2.** JPG files containing original western blots for **Figure 7B**.

of PRMT1 leads to an accumulation of intracellular lipids (*Figure 5C*). Consistently, we also observed that PRMT1 downregulates the CPT1A protein level (*Figure 7A, B*) and detected lipid accumulation (*Figure 7C*) using FACS analysis. Our unpublished data further validate that PRMT1 downregulates CPT1A in platelets expressing high levels of PRMT1.

Although PRMT1 reduces oxygen consumption and membrane potential (*Figure 4*), it does not fully inhibit the TCA cycle or the electron transport chain. Cells expressing PRMT1 (designated as 6133/PRMT1) continue to proliferate using galactose, which does not produce ATP via glycolysis in glucose-deficient medium. Moreover, the 6133/PRMT1 cells can thrive on glycerol combined with substrates such as acetate (in the form of triacetin), propionate (tripropionin), and butyrate (tributyrin), indicating that mitochondria can still utilize short-chain fatty acids (*Figure 7G*). These are converted into acetyl-CoA, propionyl-CoA, and butyryl-CoA, which can then enter the TCA cycle independently of CPT1A. This adaptation allows 6133/PRMT1 cells to consume short-chain fatty acids and non-fermentable carbon sources like glycerol and galactose in glucose-deficient medium when CPT1A is downregulated.

RBM15 also binds to several mRNAs that encode enzymes involved in glycolysis, such as LDHA and HK1. Although we did not detect increased protein expression of HK1 and LDHA, we observed an increase in the tyrosine phosphorylation of LDHA by PRMT1. Phosphorylation is known to activate LDHA, indicating that PRMT1 upregulation enhances glycolytic flux. The decreased NADP/NADPH ratio and aspartate concentration suggest a metabolic shift favoring glycolysis in proliferating cells.

The accumulation of succinate and the corresponding decrease in aspartate and alpha-ketoglutarate levels, as demonstrated by PRMT1 overexpression, is consistent with findings from earlier research involving T cells that have undergone succinate dehydrogenase (SDH) knockout (*Chen et al., 2022*). This alteration inhibits α-KG-dependent dioxygenases, including Jumonji-domain histone demethylases and DNA demethylases (such as TET), vital for the demethylation of histones and DNA. As a result, this affects chromatin accessibility and influences gene expression patterns. We propose that changes in SDH activity may allow PRMT1 to modify the epigenetic landscapes of mammalian cells indirectly in response to metabolic shifts. Given the succinate accumulation, we expect that the HIF1α complex will be stabilized, leading to pseudohypoxia phenotypes in cells with elevated PRMT1 expression, akin to those seen in patients with myelodysplastic syndromes (MDS) (*Hayashi et al., 2018*). Furthermore, our findings indicate that PRMT1 expression is significantly increased in samples from MDS patients (*Su et al., 2021*).

PRMT1-mediated metabolic reprogramming indicates that mitochondria in PRMT1-overexpressing cancer cells primarily function to produce biosynthetic precursors instead of generating oxidative ATP. Our findings further support that PRMT1 overexpression promotes mitochondrial biogenesis, as previously reported and validated here (*Figure 4C, D*). However, the increase in mitochondrial quantity did not correlate with increased oxygen consumption. Instead, we observed lower mitochondrial ROS levels per cell, consistent with reduced mitochondrial membrane potential and decreased oxygen consumption. Mitochondria play a crucial role in cancer cell metabolism, functioning not only as powerhouses for ATP production but also as central hubs for metabolic intermediates vital for biomass synthesis. While many cancer cells display a glycolytic phenotype, known as the Warburg effect, mitochondria remain essential for sustaining cell proliferation.

Intriguingly, although mitochondrial ROS levels were reduced, we detected an overall increase in cytoplasmic ROS. Since lipid accumulation can stimulate ROS generation (*Liu et al., 2015*), we speculate that intracellular lipid accumulation due to PRMT1 overexpression may drive the increased cytoplasmic ROS levels we observed in *Figure 4*. The regulation of oxidative stress by PRMT1 appears to involve a complex interplay between mitochondrial function, lipid metabolism, and glycolytic flux. In cancer cells with high PRMT1 expression, mitochondria may function as biosynthetic factories, channeling metabolic intermediates for nucleotide, amino acid, and lipid synthesis rather than oxidative phosphorylation. Understanding the precise mechanisms by which PRMT1 regulates metabolic pathways and redox homeostasis may open new therapeutic avenues for targeting metabolic vulnerabilities in cancer.

The main limitation of this study is that we used only one mouse leukemia cell line for the metabolic investigation. In our unpublished data, we have examined the expression of CPT1A in megakaryocytes and platelets derived from Pf4-cre PRMT1 transgenic mice. Additionally, we validated that PRMT1 downregulated CPT1A in the megakaryocyte lineages. Considering that RBM15 and PRMT1

are expressed in mammalian cells across different species, we suggest that the PRMT1-mediated metabolic programming is conserved.

## Materials and methods

### Cell culture and metabolite measurement

6133 cells were cultured in RPMI 1640 medium supplemented with 10% fetal bovine serum 100 U/ml penicillin and 100 µg/ml streptomycin. The addition or withdrawal of mSCF (mouse stem cell factor) (10 ng/ml) in the culture was performed accordingly. The L-Lactate Assay Kit (Cayman, Ann Arbor, Michigan) was used to measure the intracellular and extracellular lactate levels of the cultured cells: approximately $1 \times 10^7$ cells were cultured in 10 ml of fresh medium for 24 hr; the cell pellet and medium were collected and processed separately as instructed. The Glucose Colorimetric Assay Kit (Cayman, Ann Arbor, MI) was used to quantify the remaining glucose in the culture medium: $1.25 \times 10^5$ cells were cultured in fresh medium overnight; the supernatant/medium was collected and processed as instructed. The fluorescent or colorimetric signals from the assays mentioned above were measured using a microplate reader (Biotek, Winooski, VT). The ratio of NADP/NADPH was assessed using a kit (Cat# MAK479, MilliporeSigma). A total of 6133 cells (0.1 million) grown in the exponential phase were harvested for the assay in a 96-well plate according to the manufacturer's instructions. The ratio of glutathione GSH/GSSG was determined following the manufacturer's instructions (Cat# MAK440, MilliporeSigma).

### Viral production and cell line selection

For lentivirus production as described (*Su et al., 2021*), viral vectors were co-transfected with the envelope vector pMD2.G and the packaging vector ps-PAX2 into 293T cells. Fresh or concentrated viruses were then used to infect the target 6133 cell lines. Stable cell lines were selected using GFP-based flow cytometry sorting with the BD FACSAria II system (BD, Franklin Lakes, NJ).

### Murine leukemia model and treatments

GFP-positive 6133 cells were intravenously transferred into 8- to 12-week-old sub-lethally irradiated (6 Gy) C57BL/6 mice, which was approved in the IACUC APN-10182 protocol at UAB. Disease progression was closely monitored on a daily basis. Moribund mice were sacrificed to further analyze GFP-positive donor leukemia cells in the peripheral blood, bone marrow, and spleen. For PRMT1 inhibitor treatment, 80 mg/kg body weight of inhibitor solution (16 mg/ml) was intraperitoneally injected into recipient mice every other day for 1 month, starting on the fourth day post-cell transfer. The PRMT1 inhibitor was dissolved in saline with 20% Captisol, 20% PEG-400, and 5% NMP (vol/vol). For 2-DG treatment, 0.25 g/kg body weight of 2-DG (saline solution) was intraperitoneally injected into recipient mice every other day for 1 month, beginning on the fifth day post-cell transfer.

### Flow cytometry analysis

FACS analysis was conducted using the BD LSRFortessa (BD, Franklin Lakes, NJ). MitoTracker DeepRed FM and TMRE stains (Molecular Probes, Eugene, OR) were employed to evaluate cellular mitochondria. H2DCFDA dye (Cat#D399 Thermo Fisher) was used to quantify cytoplasmic ROS levels. BODIPY 493/503 (4,4-Difluoro-1,3,5,7,8-Pentamethyl-4-Bora-3a,4a-Diaza-s-Indacene) (Thermo Fisher) was added at a final concentration of 1 µM and incubated with 6133 and 6133/PRMT1 cells during the exponential phase for 15 min at 37°C in the dark. After incubation, the cells were washed with phosphate-buffered saline (PBS) twice before being subjected to FACS analysis using the manufacturer-recommended spectrum filters.

Dr. Zheng's laboratory at the University of Georgia synthesized E84, storing a 5 mM stock solution of E84 in dimethyl sulfoxide at –20°C. Exponentially growing cells were harvested and washed with ice-cold PBS. E84 was then incubated at a final concentration of 10 nM with $5 \times 10^5$ cells in 100 µl of PBS on ice for 30 min. Afterward, the cells underwent two additional washes with PBS, and the labeled cells were immediately used for cell sorting and FACS analysis. The FACS parameters included a laser wavelength of 640 nm, along with a filter set of 650 LP + 670/14. Fluorescence measurements were collected using the allophycocyanin channel on the BD LSRFortessa machine (BD, Franklin Lakes, NJ).

The FACS data were analyzed with FlowJo software, and cell sorting was performed on a BD Aria II sorter.

## SDS–PAGE and western blotting

6133 cells were collected from culture and lysed in 1 ml of H-Lysis buffer (20 mM HEPES pH 7.9, 150 mM NaCl, 1 mM $MgCl_2$, 0.5% NP40, 10 mM NaF, 0.2 mM $NaVO_4$, 10 mM β-glycerol phosphate, and 5% glycerol) with freshly added dithiothreitol (1 mM), PMSF (100 µM), and a protease inhibitor cocktail (Roche, Branford, CT). The cells were incubated on ice for 30 min and sonicated using the Bioruptor Ultra-sonication system (Diagenode, Denville, NJ). SDS–PAGE sample buffer was added to the sonicated extracts and boiled. The samples were resolved by SDS–PAGE and transferred to PVDF membranes (Millipore, Billerica, MA). The membranes were blotted with antibodies and then visualized using the Immobilon Western Chemiluminescent reagent (Millipore) with the Bio-Rad ChemiDoc MP system (Bio-Rad, Hercules, CA). The antibodies used in this study include PRMT1 (Cat# 07404, Millipore), LDHA (Cat# MA5-17246, Invitrogen), p-LDHA (Cat# 8176, Cell Signaling), and CPT1a (Cat# 66039, Proteintech).

## Quantitative real-time PCR

Total RNA was prepared using Direct-Zol RNAprep Kit (Zymo Research, Irvine, CA). cDNA was generated by the Verso cDNA synthesis Kit (Thermo Scientific, Walthum, MA) with random hexamer priming. Real-time PCR assays were performed with Absolute Blue qPCR SYBR Green Mix (Thermo Scientific) on a ViiA 7 system (Applied Biosystems, Waltham, MA). The relative quantity of gene expression was calculated by the ΔΔCt method. Housekeeping gene Actb (forward: GGC TGG CCG GGA CCT GAC AGA CTA C; reverse: GCA GTG GCC ATC TCC TGC TCG AAG TC) was used for normalization.

## Primer list

Prmt1-F CCCGTGGAGAAGGTGGACAT
Prmt1-R CTCCCACCAGTGGATCTTGT
Cpt1a-F GGCATAAACGCAGAGCATTCCTG
Cpt1a-R CAGTGTCCATCCTCTGAGTAGC
Idha-F GAATTACGATGGGGGATGTGC
Idha-R GACGTCTCTTGCCCTTTCTG
Fasn-F AAGTTCGACGCCTCCTTTTT
Fasn-R TGCCTCTGAACCACTCACAC
Murine cytochrome B forward: CTTCATGTCGGACGAGGCTTA
Murine cytochrome B reverse: TGTGGCTATGACTGCGAACA

## Cell viability assays

Cell viability was measured using the CellTiter-Glo Viability Assay Kit (Promega, Madison, WI). A total of 1000 6133 cells were seeded in 96-well plates (100 µl per well) with or without treatment. At the specified time after culture setup, 100 µl of CellTiter-Glo reagent was added to each well. The luminescent signal was recorded with a microplate reader (Biotek, Winooski, VT).

## Metabolite extraction

Cells were washed twice with ice-cold PBS before extracting metabolites in a 20% methanol solution (LC–MS grade methanol, Fisher Scientific) at −70°C. The tissue–methanol mixture underwent bead-beating for 45 s using a Tissue/cell disruptor (QIAGEN). The extracts were centrifuged for 5 min at $2000 \times g$ to pellet insoluble material, and the supernatants were transferred to clean tubes. This extraction procedure was repeated two more times, and all three supernatants were pooled, dried using a Vacufuge (Eppendorf), and stored at −80°C until analysis. The methanol-insoluble protein pellet was solubilized in 0.2 M NaOH at 95°C for 20 min, and the total protein concentration was quantified using a Bio-Rad DC assay. On the day of metabolite analysis, the dried cell extracts were reconstituted in 70% acetonitrile to achieve a relative protein concentration of 1 µg/ml, and 4 µl of this reconstituted extract was injected for LC/MS-based untargeted metabolite profiling.

## LC/MS metabolomics

Cell extracts were analyzed using an LC/MS system that featured an Agilent Model 1290 Infinity II liquid chromatography platform coupled with an Agilent 6550 iFunnel time-of-flight mass spectrometer. The chromatography of metabolites utilized aqueous normal phase chromatography on a Diamond Hydride column (Microsolv). The mobile phases consisted of (A) 50% isopropanol with 0.025% acetic acid and (B) 90% acetonitrile containing 5 mM ammonium acetate. To minimize the interference of metal ions on chromatographic peak integrity and electrospray ionization, EDTA was introduced to the mobile phase at a final concentration of 6 µM. The mobile phase gradient was as follows: 0–1.0 min, 99% B; 1.0–15.0 min, decreasing to 20% B; 15.0–29.0 min, 0% B; and from 29.1 to 37 min, returning to 99% B. Raw data were analyzed with MassHunter Profinder 8.0 and MassProfiler Professional (MPP) 15.1 software (Agilent Technologies).

## Metabolite structure specification

To determine the identities of differentially expressed metabolites ($p < 0.05$), LC/MS data were queried against an in-house annotated personal metabolite database, created using MassHunter PCDL Manager 8.0 (Agilent Technologies), based on monoisotopic neutral mass (<5 ppm mass accuracy) and chromatographic retention times of pure standards. A molecular formula generator (MFG) algorithm in MPP was utilized to generate and score empirical molecular formulas, considering monoisotopic mass accuracy, isotope abundance ratios, and spacing between isotope peaks. A tentative compound ID was assigned when the PCDL database and MFG scores matched for a given candidate molecule. Tentatively assigned molecules were confirmed based on matching LC retention times and/or MS/MS fragmentation spectra for pure molecular standards.

## Seahorse assays

The metabolic activities of living 6133 cells and 6133/PRMT1 cells were measured using a Seahorse XF24 metabolic flux analyzer (Agilent Technologies, Santa Clara, CA), following the described procedures (*Yu et al., 2013*). Briefly, AMKL cells ($1.5–2 \times 10^5$) were seeded in 24-well plates designed for XF24 cell culture. The plates were pre-coated with Cell-Tak (Corning, Corning, NY) and incubated in XF base medium containing glucose (10 mM), L-glutamine (2 mM), and sodium pyruvate (1 mM) at 37°C for 1 hr. OCRs were measured at baseline and after the sequential addition of the mitochondrial inhibitor oligomycin (500 nM), the mitochondrial uncoupling compound FCCP (5 µM), and the respiratory chain inhibitor antimycin (1 µM) to the culture. The FCCP concentration must be determined by titration. Glycolytic activities were simultaneously assessed using the same instrument based on ECARs. All measurements were conducted according to the manufacturer's protocols, and the data were normalized by cell numbers.

## Statistics and data analysis

The figure legends provide all the experimental details. In the bar graphs, a two-tailed Student's *t*-test was employed for significance testing, with p values less than 0.05 deemed significant. Quantitative data are presented as mean ± SEM. The R programming package and GraphPad Prism 6 were utilized for statistical analysis.

# Acknowledgements

The project was partially supported by the Leukemia Research Foundation, NCI R21 CA202390, and the Elsa Pardee Foundation. ML is supported by the NIH R35 grant GM131858. YC is supported by NIH grants HL146103, HL158097, HL167201, and AG082839, as well as the United States Department of Veterans Affairs research awards BX005800, BX004426, BX006321, and CX002706 (to YC). We would like to thank Dr Taro Hito Hitosugi (Mayo Clinic) for the CPT1A cDNAs.

# Additional information

## Funding

| Funder | Grant reference number | Author |
|---|---|---|
| Leukemia Research Foundation | | Xinyang Zhao |
| National Cancer Institute | R21 CA202390 | Xinyang Zhao |
| Elsa Pardee Foundation | | Xinyang Zhao |
| National Institutes of Health | R35 grant GM131858 | Minkui Luo |
| National Institutes of Health | HL146103 | Yabing Chen |
| U.S. Department of Veterans Affairs | BX005800 | Yabing Chen |
| National Institutes of Health | HL158097 | Yabing Chen |
| National Institutes of Health | HL167201 | Yabing Chen |
| National Institutes of Health | AG082839 | Yabing Chen |
| U.S. Department of Veterans Affairs | BX004426 | Yabing Chen |
| U.S. Department of Veterans Affairs | BX006321 | Yabing Chen |
| U.S. Department of Veterans Affairs | CX002706 | Yabing Chen |

The funders had no role in study design, data collection, and interpretation, or the decision to submit the work for publication.

## Author contributions

Hairui Su, Conceptualization, Resources, Data curation, Formal analysis, Validation, Investigation, Methodology, Writing – original draft; Yong Sun, Han Guo, Data curation, Investigation; Chiao-Wang Sun, Glen Raffel, Investigation, Methodology; Qiuying Chen, Data curation; Szumam Liu, Investigation; Anlun Li, Validation, Visualization; Min Gao, Software; Rui Zhao, Data curation, Formal analysis; Jian Jin, Cheng-Kui Qu, Resources; Michael Yu, Data curation, Supervision; Christopher A Klug, Scott Ballinger, Long X Zheng, Steven Gross, Supervision; George Y Zheng, Resources, Validation; Matthew Kutny, Zechen Chong, Resources, Supervision; Chamara Senevirathne, Resources, Validation, Investigation; Yabing Chen, Conceptualization, Resources, Data curation, Supervision, Investigation, Methodology, Writing – review and editing; Minkui Luo, Conceptualization, Resources, Data curation, Software, Formal analysis, Supervision, Investigation, Writing – review and editing; Xinyang Zhao, Conceptualization, Data curation, Formal analysis, Supervision, Investigation, Methodology, Writing – original draft, Project administration, Writing – review and editing

## Author ORCIDs

Yong Sun 
Qiuying Chen 
Jian Jin 
Cheng-Kui Qu 
George Y Zheng 
Long X Zheng 
Minkui Luo 
Xinyang Zhao 

Reviewer #1 (Public review): https://doi.org/10.7554/eLife.105318.3.sa1
Reviewer #2 (Public review): https://doi.org/10.7554/eLife.105318.3.sa2
Author response https://doi.org/10.7554/eLife.105318.3.sa3

## Additional files

### Supplementary files
MDAR checklist

### Data availability
All data generated or analyzed during this study are included in the manuscript and supporting files.

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
