## [Editor Report · eLife Assessment]

This study reveals that PRMT1 overexpression drives tumorigenesis of acute megakaryocytic leukemia (AMKL) and that targeting PRMT1 is a viable approach for treating AMKL. After revision, both reviewers found that these findings are **important** and that the data supporting these findings are **convincing**. Furthermore, these findings likely have significant implications for the treatment of AMKL with PRMT1 overexpression in the future.

---

## [Referee Report · Reviewer #1 (Public review)]

Summary:

PRMT1 overexpression is linked to poor survival in cancers, including acute megakaryocytic leukemia (AMKL). This manuscript describes the important role of PRMT1 in the metabolic reprograming in AMKL. In a PRMT1-driven AMKL model, only cells with high PRMT1 expression induced leukemia, which was effectively treated with the PRMT1 inhibitor MS023. PRMT1 increased glycolysis, leading to elevated glucose consumption, lactic acid accumulation, and lipid buildup while downregulating CPT1A, a key regulator of fatty acid oxidation. Treatment with 2-deoxy-glucose (2-DG) delayed leukemia progression and induced cell differentiation, while CPT1A overexpression rescued cell proliferation under glucose deprivation. Thus, PRMT1 enhances AMKL cell proliferation by promoting glycolysis and suppressing fatty acid oxidation.

Strengths:

This study highlights the clinical relevance of PRMT1 overexpression with AMKL, identifying it as a promising therapeutic target. A key novel finding is the discovery that only AMKL cells with high PRMT1 expression drive leukemogenesis, and this PRMT1-driven leukemia can be effectively treated with the PRMT1 inhibitor MS023. The work provides significant metabolic insights, showing that PRMT1 enhances glycolysis, suppresses fatty acid oxidation, downregulates CPT1A, and promotes lipid accumulation, which collectively drive leukemia cell proliferation. The successful use of the glucose analogue 2-deoxy-glucose (2-DG) to delay AMKL progression and induce cell differentiation underscores the therapeutic potential of targeting PRMT1-related metabolic pathways. Furthermore, the rescue experiment with ectopic Cpt1a expression strengthens the mechanistic link between PRMT1 and metabolic reprogramming. The study employs robust methodologies, including Seahorse analysis, metabolomics, FACS analysis, and in vivo transplantation models, providing comprehensive and well-supported findings. Overall, this work not only deepens our understanding of PRMT1's role in leukemia progression but also opens new avenues for targeting metabolic pathways in cancer therapy.

Comments on revisions:

The reviewer's questions were adequately addressed.

---

## [Referee Report · Reviewer #2 (Public review)]

Summary:

The manuscript explores the role of PRMT1 in AMKL, highlighting its overexpression as a driver of metabolic reprogramming. PRMT1 overexpression enhances the glycolytic phenotype and extracellular acidification by increasing lactate production in AMKL cells. Treatment with the PRMT1 inhibitor MS023 significantly reduces AMKL cell viability and improves survival in tumor-bearing mice. Intriguingly, PRMT1 overexpression also increases mitochondrial number and mtDNA content. High PRMT1-expressing cells demonstrate the ability to utilize alternative energy sources dependent on mitochondrial energetics, in contrast to parental cells with lower PRMT1 levels.

Strengths:

This is a conceptually novel and important finding as PRMT1 has never been shown to enhance glycolysis in AMKL, and provides a novel point of therapeutic intervention for AMKL.

Comments on revisions:

The author has responded satisfactorily to the review comments and revised the manuscript accordingly.

---

## [Author Response]

The following is the authors’ response to the original reviews

**Reviewer #1:**

We thank the reviewer for highlighting the strength in our manuscript as quote: “Overall, this work not only deepens our understanding of PRMT1's role in leukemia progression but also opens new avenues for targeting metabolic pathways in cancer therapy.”

Weakness :(1) The findings rely heavily on a single AMKL cell line, with no validation in patient-derived samples to confirm clinical relevance or even another type of leukemia line. Adding the discussion of PRMT1's role in other leukemia types will increase the impact of this work.

We mentioned in the introduction that PRMT1 is known to be the driver for leukemia with diverse types of mutations. In a related paper published in Cell Reports (Su et al. 2021), we demonstrated that PRMT1 is upregulated in MDS myeloid dysplasia syndrome patient samples and that the inhibition of PRMT1 promotes megakaryocytic differentiation of a few MDS samples. AMKL is very rare. Via Children’s Oncology group consortium, we have obtained five AMKL samples with Down’s syndrome and AMKL with RBM15-MKL1 translocation out of 32 samples in the bank over the last 20 years. Interestingly, these patient samples also contain trisomy 19. As PRMT1 is localized on chromosome 19, we speculate that PRMT1 is the significant driver for AMKL leukemia, although we have very limited genetic evidence. However, these human frozen samples derived from peripheral blood cannot be grown in a cell culture system. Although we did not perform metabolic analysis for other AMKL cell lines, we did validate in our unpublished studies that PRMT1 drives down CPT1A expression in normal bone marrow cells and platelets in mice and in human leukemia cell line called MEG-01, which can be differentiated into megakaryocytes upon PMA (phorbol 12-myristate 13-acetate) treatment. Therefore, we expect that the PRMT1-mediated metabolic reprogramming we described here should apply to other types of hematological malignancies.

(2) The observed heterogeneity in Prmt1 expression is noted but not further investigated, leaving gaps in understanding its broader implications.

The expression level of PRMT1 is heterogeneous within leukemia cell populations, making it intriguing to study. We can sort the cells based on high versus low PRMT1 expression using a fluorescent dye called E84. However, we have not conducted transcriptome analysis on these two populations, mainly due to resource constraints. Theoretically, the E84 high-expression population may transiently utilize glucose more efficiently, as these cells do not ectopically express PRMT1. Therefore, when nutrient levels decline, these cells might switch to the low PRMT1 expression population. It will be interesting to see whether endogenous leukemia cells transiently expressing high levels of PRMT1 take advantage of their efficient usage of glucose and thus adapt to the niche environment successfully, as we observed in the Figure 1. I agree that this would be an interesting direction to pursue in the future.

(3) Some figures and figure legends didn't include important details or had not matching information.

We would like to thank the reviewer for pointing out these mistakes. Now we have corrected.

(4) Some wording is not accurate, such as line 80 "the elevated level of PRMT1 maintains the leukemic stem cells", the study is using the cell line, not leukemia stem cells.

Leukemic stem cells are often referred to as cells that can initiate leukemia when transplanted into recipient mice, a concept first proposed by John Dick. In this study, we found that even the 6133 cell line displays heterogeneity in terms of PRMT1 expression levels. We identified a subgroup of 6133 cells as leukemia stem cells due to their ability to initiate leukemia.

(5) In the disease model, histopathology of blood, spleen, and BM should be shown.

We did not conduct histopathology analysis. 6133 cells associated histopathology has been published in Mercher et al JCI 2009 and a recent preprint by Diane Krause’s group.

(6) Can MS023 treatment reverse the metabolic changes in PRMT1 overexpression AMKL cells?

Yes, We demonstrated in figure 4 in the seahorse assays that prmt1 inhibitor can increase the oxygen consumption.

It would be helpful to provide a summary graph at the end of the manuscript.

Yes, we now provide a graphic abstract.

**Reviewer #2 (Public review):**

We would like to thank the reviewer for finding the manuscript novel and important.

Weaknesses:(1) The manuscript lacks detailed molecular mechanisms underlying PRMT1 overexpression, particularly its role in enhancing survival and metabolic reprogramming via upregulated glycolysis and diminished oxidative phosphorylation (OxPhos). The findings primarily report phenomena without exploring the reasons behind these changes.

In the introduction, we highlighted that numerous studies have demonstrated how PMT1 directly interacts with several key enzymes involved in glycolysis. These studies provide a mechanism for the observed upregulation of PMT1 in leukemia. Additionally, our previous research published in eLife 2015 {Zhang, 2015 #5031} demonstrated that PRMT1 methylates the RNA-binding protein RBM15, which can bind to the 3' UTR of mRNAs encoding various metabolic enzymes. Therefore, we propose that PMT1 may also regulate metabolism indirectly through the RBM15 protein.

(2) The article shows that PRMT1 overexpression leads to augmented glycolysis and low reliance on the OxPhos. However, the manuscript also shows that PMRT1 overexpression leads to increased mitochondrial number and mitochondrial DNA content and has an elevated NADPH/NAD+ ratio. Further, these overexpressing cells have the ability to better survive on alternative energy sources in the absence of glucose compared to low PMRT1-expressing parental cells. Surprisingly, the seashores assay in PRMT1 overexpressing cells showed no further enhancement in the ECAR after adding mitochondrial decoupler FCCP, indicating the truncated mitochondrial energetics. These results are contradicting and need a more detailed explanation in the discussion.

We have explained the metabolic changes in more detail now. Increasing mitochondria number is not equivalent to increasing fatty acid oxidation and oxygen consumption, as the mitochondria have many other functions. PRMT1 only downregulates CPT1A, which is a rate-limiting step for long-chain fatty acid oxidation. The data suggest that PRMT1 promotes the biogenesis of mitochondria maybe via PGC1alpha as published by Stallcup’s group. The seahorse assays were performed in the high concentration of glucose instead of alternative carbon sources. FCCP treatment under high glucose conditions did not increase the ECR and OCR, which is normal for leukemia cells as shown in other people’s publications {Sriskanthadevan, 2015 #3944}{Kreitz, 2019 #2133}. PRMT1 could dampen the activities of TCA cycle and the electron transportation chain as the proteomic data from our unpublished data and published data {Fong, 2019 #1185} suggested. The elevated NADPH/NAD+ ratio is another indication that glycolysis and anabolism are enhanced by PRMT1.

(3) How was disease penetrance established following the 6133/PRMT1 transplant before MS023 treatment?

Yes, the data was in figure 1f, demonstrating that the penetrance is 100%.

(4) The 6133/PRMT1 cells show elevated glycolysis compared to parental 6133; why did the author choose the 6133 cells for treatment with the MS023 and ECAR assay (Fig.3 b)? The same is confusing with OCR after inhibitor treatment in 6133 cells; the figure legend and results section description are inconsistent.

Sorry for the mistakes while we are preparing the manuscript. We used 6133/PRMT1 cells to be treated with MS023 in figure 4.

(5) The discussion is too brief and incoherent and does not adequately address key findings. A comprehensive rewrite is necessary to improve coherence and depth.

We agree with the reviewer. Now we added comprehensive review of PRMT1-mediated metabolism. The PRMT1 homolgous in yeast is called hmt1. In yeast, hmt1 is upregulated by glucose and enhance glycolysis. So PRMT1 enhanced glycolysis is a conserved pathway in eukaryocytic cells.

(6) The materials and methods section lacks a description of statistical analysis, and significance is not indicated in several figures (e.g., Figures 1C, D, F; Figures 2D, E, F, I). Statistical significance must be consistently indicated. The methods section requires more detailed descriptions to enable replication of the study's findings.

We have added extra details on the methods and statistical analysis for the figures.

(7) Figures are hazy and unclear. They should be replaced with high-resolution images, ensuring legible text and data.

We have prepared separate figure files with high resolution.

(8) Correct the labeling in Figure 2I by removing the redundant "D."

We would like to thank the reviewer and fixed the figure.